# The Integration of Patient-Centered Care and the Biopsychosocial Model by Athletic Trainers in the Secondary School Setting

**DOI:** 10.3390/ijerph20085480

**Published:** 2023-04-12

**Authors:** Adriana M. Mendoza, Matthew J. Drescher, Lindsey E. Eberman

**Affiliations:** Department of Applied Medicine and Rehabilitation, Indiana State University, Terre Haute, IN 47803, USA

**Keywords:** patient-centered care, biopsychosocial, whole-person healthcare, healthcare competency

## Abstract

Our purpose was to explore the degree to which secondary school athletic trainers (SSATs) perceive they are integrating the principles of patient-centered care (PCC) and the biopsychosocial (BPS) model in their practice. We used a cross-sectional design to explore the primary research question. We used the Global Perceptions of Athletic Trainer Patient-Centered Care (GPATPCC) tool and the Biopsychosocial Model of Health (BPSMH) tool, both measured on a 4-point Likert scale (1 = strongly disagree, 2 = disagree, 3 = agree, 4 = strongly agree, with an unscored “unsure” option). We sent the survey to 5665 SSATs through the National Athletic Trainers’ Association. Results indicate participants expressed strong agreement (mode = 4) with 7 of the 14 statements and agreement (mode = 3) with the remaining 7 statements of the GPATPCC tool (grand mean = 3.4 ± 0.8). Overall, participants rated their level of agreement on the BPSMH as agreeing (mode = 3) for each item (grand mean = 3.0 ± 1.0). SSATs perceive they are integrating the principles of PCC and the BPS model in clinical practice. These findings align with two previous studies concluding that patients, parents, and providers believe athletic trainers provide care that is focused on whole-person healthcare.

## 1. Introduction

Patient-centered care (PCC) has become a fundamental component of health care delivery, characterized by responsiveness to and respect for the patient’s preferences, needs, and values, and ensuring that their values guide all clinical decisions [1,2]. The 8 domains of PCC are outlined in Table 1 [2]. PCC has been recognized as a crucial component of healthcare delivery in various professions, including athletic training [3]. Athletic trainers must be able to communicate and integrate the vital elements of PCC into their practice to create and promote an evidence-based care environment for their patients [4]. Social and contextual factors, such as a patient’s social network, family, community, and larger social spheres of influence, have previously been overlooked in sports medicine, but they have a direct impact on their health [5,6]. These factors depict the social determinants of health (SDOH), which are the conditions in which individuals are born, grow, live, work, and age [7]. The 5 domains of the SDOH include economic stability, education access and quality, healthcare access and quality, neighborhoods and built environments, and social and community context [8], which affect every stage of life, including early childhood and adolescence [9]. SDOH influences patient health outcomes, and the negative impact of these factors leads to decreased patient health [10,11]. Athletic trainers must recognize patients as whole human beings situated within the social context of sport to foster an environment where psychological and social factors are examined alongside the biological [5]. Patient outcomes improve by examining social and contextual factors, as these factors reveal that injury and illness are more than biological issues [12]. When all aspects of health are considered, a root cause can be determined, which can lead to actionable change [13]. The biopsychosocial (BPS) model of health is a framework that identifies and recognizes the biological, psychological, and social factors that influence a patient’s health [14,15]. This framework can be beneficial in practice as it allows patients to be treated from a disease standpoint but also creates intentional space for psychological and social information in the care process [15]. Health is a dynamic process of well-being, and when patients are evaluated through a BPS lens, this provides the opportunity to integrate psychological and social factors that play a significant role in individual lives [12]. 

However, while medicine has studied the integration of PCC and the BPS in patient care, there has been little to no research on the integration of PCC and the BPS model in athletic training, specifically at the secondary school level. The purpose of our study was to explore the degree to which secondary school athletic trainers (SSATs) perceive they are integrating the principles of PCC and the BPS model in their practice. We hypothesize that athletic trainers in the secondary school believe they are integrating the principles of PCC and the BPS model into their practice.

## 2. Materials and Methods

### 2.1. Study Design 

We used a cross-sectional research design to explore the primary research question. This study was deemed exempt by the XXX Institutional Review Board. 

### 2.2. Participants 

We recruited SSATs from the National Athletic Trainers’ Association through the Research Survey Service. In total, 5667 emails were sent, and 45 were undeliverable; thus, 5622 emails were successfully sent. A total of 422 SSATs accessed the survey (access rate = 7.5%). Of those who accessed the survey, 18 participants either did not consent to participate in the study or did not complete the survey. Thirteen participants were excluded from the final analysis because each identified that they do not currently practice in the secondary school setting. A total of 404 SSATs were used for the analysis (completion rate = 95.7%). Participants mostly identified as women (*n* = 215, 61.3%) and indicated they worked full-time at their secondary school (*n* = 249, 70.9%). Full demographic information is presented in Table 2. 

### 2.3. Instrumentation

We constructed a survey to assess SSATs’ perceptions of providing PCC and integrating the BPS model with patients. This survey is a combination of two validated tools, with slight modifications made to each. The survey consisted of eight demographic questions followed by the Global Perceptions of Athletic Trainer Patient-Centered Care (GPATPCC) tool (Table 3) [1] and the BPS Model of Health tool (Table 4). The pre-existing GPATPCC tool’s wording was slightly edited to reflect the point of view of the athletic trainer instead of the student-athlete, for whom it was initially created [1]. The tool contains 15 items measured on a 4-point Likert scale (1 = strongly disagree, 2 = disagree, 3 = agree, 4 = strongly agree, with an unscored “unsure” option) that ask athletic trainers to rate their level of agreement with statements regarding how they felt they provided PCC to their patients [1]. This tool was designed from the Picker 8 domains and principles, was reviewed for accuracy and reliability by an external expert in PCC relative to athletic training, and has strong internal consistency (Cronbach α 0.897) [1]. The Biopsychosocial Model of Health tool contains three items about the athletic trainer’s perceptions of integrating the BPS model of health into their clinical practice, measured on a 4-point Likert scale (1 = strongly disagree, 2 = disagree, 3 = agree, 4 = strongly agree, with an unscored “unsure” option). The tool was constructed by the research team using the three constructs of the BPS model: (1) biological influences, (2) psychological influences, and (3) social influences [4]. After the tool was developed, it was sent out for external review. Reviewers were selected for their experience as providers and researchers in athletic training and physical therapy. Three reviewers provided feedback on the relevance (1 = not relevant, 2 = item needs some revision, 3 = relevant but needs minor revision, 4 = very relevant) and clarity (1 = not clear, 2 = item needs some revision, 3 *=* clear but needs minor revision, 4 = very clear) of items. The tool was sent through one round of content analysis, where the reviewers scored each item and provided qualitative feedback for clarity, leading to minor editorial changes. The tool’s calculated content validity index (CVI) is a scale-level content validity index based on the universal agreement (S-CVI/UA) = 1.00. 

### 2.4. Procedures 

Emails were sent to SSATs describing the study and providing directions for completion, along with a direct link to access the survey via a secure, web-based system (Qualtrics, Provo, UT, USA). Once participants accessed the survey and indicated they currently work in a secondary school setting, they were taken to the electronic informed consent form. After agreeing to participate in the study, they were directed to the demographic questions as well as the GPATPCC and the BPS Model of Health tools. The survey was open for two 3-week data collection periods, one in April 2022 and one in September 2022, for a total of six weeks. The data collection period in September 2022 was extended by an additional two weeks to allow adequate time for participants to complete the survey. 

### 2.5. Data Analysis 

Descriptive statistics (mean, SD, mode, and frequency) were used to characterize the data. In light of some literature suggesting there may be gender differences in the delivery of patient-centered care [16,17], we compared men and women on their responses to the GPATPCC and BPS Models of Health using separate Mann-Whitney U tests. Significance was identified if *p* < 0.05. 

## 3. Results

### 3.1. GPATPCC Tool 

Participants expressed strong agreement (mode = 4) with seven of the 14 statements and agreement (mode = 3) with the remaining seven statements of the GPATPCC tool (grand mean = 3.4 ± 0.8) (Table 3). The two highest-ranked statements indicated that the SSATs felt they have not made their patients participate in competition when deemed “medically out of participation” (mean = 3.7 ± 0.6) and that they have informed their patients of their clinical status (mean = 3.6 ± 0.6). However, on average, the two lowest-ranked statements indicated that the SSATs felt they have addressed the patient’s access to care, including transportation, ease of scheduling, and accessibility to specialist referral (mean = 2.8 ± 1.1), and that they are able to recognize any conflict of interest that could adversely affect the patient’s health (mean = 2.9 ± 1.2). Only 1.7% (*n* = 6) of participants strongly disagreed that they provide culturally competent care for their patients, and 1.1% (*n* = 4) of participants strongly disagreed that they have not made their patients participate in competition when deemed “medically out of participation”.

Significant differences were identified between genders on one statement: “address my patient’s fears and anxieties regarding their clinical status, the financial effect of the injury, and the effect of their condition on others” (Mann Whitney U = 12,320, z = −2.427, *p* = 0.015). On average, women (mean = 3.4 ± 0.8) rated themselves 8.3% higher than men respondents (mean = 3.1 ± 1.2).

**Table 3 ijerph-20-05480-t003:** Global Perceptions of Athletic Trainer Patient-Centered Care Tool Items and Responses ^a^.

Prompt: As an Athletic Trainer in the Secondary School Setting, I Feel I…	Score ^b^	No. of Participants n/N, %				
	Mean ± SD (Mode)	Strongly Disagree	Disagree	Agree	Strongly Agree	Unsure
Provide culturally competent care for patients.	3.5 ± 0.8 (4)	6/351, 1.7	0/351, 0.0	143/351, 40.7	196/351, 55.8	6/351, 1.7
Deliver care that is respectful of my patients’ preferences.	3.5 ± 0.6 (4)	3/351, 0.9	1/351, 0.3	150/351, 42.7	194/351, 55.3	3/351, 0.9
Provide care that is respectful of the patient’s preferences.	3.4 ± 0.8 (4)	3/351, 0.9	1/351, 0.3	165/351, 47.0	173/351, 49.3	9/351, 2.6
Inform my patients of their clinical status.	3.6 ± 0.6 (4)	2/351, 0.6	0/351, 0.0	134/351, 38.2	212/351, 60.4	3/351, 0.9
Promote a healthy lifestyle for my patients.	3.4 ± 0.7 (4)	3/351, 0.9	1/351, 0.3	166/351, 47.3	174/351, 49.6	7/351, 2.0
Provide education and information to patients.	3.5 ± 0.6 (3)	1/351, 0.3	0/351, 0.0	176/351, 50.1	171/351, 48.7	3/351, 0.9
▪ Address my patients’ pain, ADLs, and environment.	3.4 ± 0.7 (3)	1/350, 0.3	2/350, 0.6	184/350, 52.6	155/350, 44.3	8/350, 2.3
Recognize any conflicts of interest that could impact patients.	2.9 ± 1.2 (3)	1/351, 0.3	2/351, 0.6	193/351, 55.0	113/351, 32.2	42/350, 12.0
Coordinate other care for my patients.	3.1 ± 0.9 (3)	2/351, 0.6	18/351, 5.1	194/351, 55.3	119/351, 33.9	18/351, 5.1
▪ Have not made my patients participate competition when deemed “medically out of participation”.	3.7 ± 0.6 (4)	4/351, 1.1	4/351, 1.1	79/351, 22.5	261/351, 74.4	3/351, 0.9
▪ Address my patient’s access to care.	2.8 ± 1.1 (3)	3/351, 0.9	33/351, 9.4	199/351, 56.7	81/351, 23.1	35/351, 10.0
▪ Support inclusion of friends and family in decision-making.	3.4 ± 0.8 (3)	1/351, 0.3	1/351, 0.3	174/351, 49.6	166/351, 47.3	9/351, 2.6
▪ Make decisions on patient care without influence from coaches.	3.5 ± 0.8 (4)	1/351, 0.3	3/351, 0.9	140/351, 39.9	198/351, 56.4	9/351, 2.6
▪ Address my patient’s potential fears and anxieties.	3.3 ± 1.0 (3)	3/351, 0.9	4/351, 1.1	170/351, 48.4	155/351, 44.2	19/351, 5.4

^a^ Items were abbreviated from their original format. ^b^ Responses were scored as 1 (strongly disagree), 2 (disagree), 3 (agree), or 4 (strongly agree), with an unscored “unsure” option.

### 3.2. BPS Model of Health Tool 

Overall, on the BPS Model of Health tool, participants rated their level of agreement as *agree* for each item (grand mean = 3.0 ± 1.0) (Table 4). The highest-ranked statement was that participants felt they integrate the biological influences on health into their patient care (mean = 3.1 ± 1.0). In contrast, the remaining lowest-ranked statements were that participants felt they integrate psychological influences (mean = 3.0 ± 1.1) and social influences (mean = 3.0 ± 1.1) into their patient care.

Significant differences emerged between men and women on their ratings for psychological (Mann Whitney U = 12,308.5, z = −2.467, *p* = 0.014) and social influences (Mann Whitney U = 12,364.5, z = −2.445, *p* = 0.014). Again, women rated themselves higher for both the integration of psychological (women = 3.1 ± 1.0; men = 2.8 ± 1.2) and social influences (women = 3.2 ± 0.8; men = 2.8 ± 1.3), 10.5% and 13.1%, respectively.

**Table 4 ijerph-20-05480-t004:** Biopsychosocial Model of Health Tool Items and Responses ^a^.

Prompt: As an Athletic Trainer in the Secondary School Setting, I Feel I…	Score ^b^	No. of Participants n/N, %				
	Mean ± SD (Mode)	Strongly Disagree	Disagree	Agree	Strongly Agree	Unsure
Integrate the biological influences on health into my patient’s care.	3.1 ± 1.0 (3)	1/351, 0.3	4/351, 1.1	215/351, 61.3	107/351, 30.5	24/351, 6.8
Integrate the psychological influences on health into my patient’s care.	3.0 ± 1.1 (3)	1/351, 0.3	5/351, 1.4	200/351, 57.0	111/351, 31.6	34/351, 9.7
Integrate the social influences on health into my patient’s care.	3.0 ± 1.1 (3)	1/351, 0.3	5/351, 1.4	211/351, 60.1	104/351, 29.6	30/351, 8.5

^a^ Items were abbreviated from their original format. ^b^ Responses were scored as 1 (strongly disagree), 2 (disagree), 3 (agree), or 4 (strongly agree), with an unscored “unsure” option.

## 4. Discussion

The purpose of this study was to explore the degree to which SSATs perceive that they are integrating the principles of PCC and the BPS model into their practice. Our results indicate that SSATs perceive they are effective in integrating the principles of PCC and the BPS model of health into their practice. These outcomes, specifically the PCC outcomes, align with two previous studies that concluded collegiate student-athletes and parents of minor student-athletes perceive athletic trainers as providing PCC [1,4]. Athletic trainers frequently serve as primary healthcare providers in the education system [4], with 66% of secondary schools providing some degree of access to athletic training services [18]. The highest-ranked items on the GPTATPCC tool relate to SSATs not making their patients participate when they are unable to and informing their patients of their clinical status. Similarly, the highest-ranked item on the BPSMH tool was the SSATs ability to integrate the biological influences of health into patient care, closely followed by social and psychological influences. Moreover, there are significant differences between men and women, specifically relative to integrating the psychological and social influences of the BPS model. 

These findings are encouraging, as athletic trainers need to implement PCC and a BPS approach to patient care, particularly as part of a greater system in the secondary school setting. School-based health centers (SBHC) are an effective multi-sector approach in which public health intervention strategies can be implemented to address the SDOH within the adolescent patient population [19,20,21]. SBHCs, like secondary schools, often include athletic trainers and school nurses, both of whom work closely with adolescent patients and should be able to advocate for and address the various factors affecting them [19]. 

### 4.1. Strategies for Implementation 

Athletic training education is guided by the Commission on Accreditation of Athletic Training Education (CAATE) professional program standards. Although the recently updated 2020 CAATE professional program standards incorporate a focus on PCC principles and the SDOH in professional education, traditional athletic training education has had little focus on the SDOH, even though athletic trainers handle such factors and influences in daily practice [10]. Current practicing clinicians may not have received direct education on PCC or SDOH, an education that would encompass a BPS approach to healthcare [10,22]. A shift in focus within education and research toward SDOH may have a greater impact on patient and population health outcomes [10]. Although the results of this study demonstrated that SSATs perceive they are integrating the principles of PCC and the BPS model of health into their clinical practice, there is still room for improvement in incorporating these concepts, specifically the psychological and social aspects of the BPS among men SSATs. The first step for SSATs is to acknowledge that SDOH may be contributing to patient outcomes and then aim to ‘better understand the complexity of their interactions to provide culturally competent care to student-athletes [10]. However, providers in the secondary school setting have described feeling unsure how to discuss SDOH with their patients [23,24], as these conversations can be sensitive and difficult to have. A few concerns described in the previous literature are that providers feel that SDOH conversations could be uncomfortable for patients to discuss, they don’t want to jeopardize patient relationships, or they lack a way to initiate the conversation with their patients [21,25,26]. This could potentially lead to SSATs prioritizing the biological factors more frequently in their patient care than the psychosocial and social influences, even if just implicitly. One intervention that could benefit athletic trainers in the secondary school setting is a script designed to facilitate conversations between clinicians and adolescents related to SDOH [21]. If athletic trainers can initiate the discussion of SDOH, they can then assess the extent to which their patients are affected by them, thus leading to more effective care plans specifically tailored to individual patient needs [21]. 

Another strategy that could encourage and promote patient-centeredness is the use of health informatics. Information is critical to patient care, and health informatics focuses on how that information is acquired, stored, and used [27]. Athletic trainers may use tools such as electronic health records (EHRs) or electronic patient-reported outcome (ePRO) assessments to do this. EHRs are regularly used to store and transfer patient data. Still, they could potentially be more effective when used at the point of care to educate patients, engage in the co-creation of notes, and use them to ensure the accuracy of data being entered into the EHR [27]. Informatics is a growing field within health care, with patients consistently exposed to information, whether it is high-quality or misinformed. Athletic trainers and the systems in which they work should strive for patient empowerment through electronic systems to promote communication [27]. Further, EHRs have the potential to facilitate better relationships between providers to advance patient care. Routine patient-reported outcomes have indicated improvements in patient care by improving communication [27] and encouraging the patient to become an active participant in their health. ePROs provide flexibility and ease of use that may be useful in the secondary school setting [27]. Informatics is changing the way providers and patients interact, and the benefits include ‘better access to health information and health services, improved patient care and safety, greater coordination of care, and more empowered patients’ [27].

Implementing PCC into practice requires a multidimensional approach within systems, with a focus on applying a BPS perspective. In the case of one healthcare system, they curated a framework of three dimensions to enable them to be more patient-centered, consisting of an interpersonal dimension, a clinical dimension, and a structural dimension [28]. Within the interpersonal dimension, the focus lies on communication, knowing the patient, and acknowledging that all members of a system affect the system’s relationship with the patient [28]. This is important, particularly in the secondary school setting, as patients may interact with a multitude of providers. The core of the interpersonal dimension is effective communication, and ways in which systems can better work towards PCC include improving training programs among providers, enhancing workshops, surveying patients, etc. [28]. The clinical dimension includes decision support, coordination, care management, and continuity [28]. The focus of this dimension is equipping patients to manage their injuries or illnesses outside of the athletic training facility. This requires athletic trainers to leverage healthcare information technology, guide patients to high-quality information and resources, and provide support and appropriate referrals [28]. The final dimension emphasizes the importance of the built environment in which patients receive care [28]. Structuring athletic training facilities to enhance the patient experience and create a trusting environment provides them with the opportunity to seek care. These three dimensions can be used as a foundation for athletic trainers to make ‘patient-centered care part of the culture of care’ [28] within their systems. 

### 4.2. Limitations and Future Research 

In this study, we explored the perceptions of SSATs related to integrating the principles of PCC and the BPS model of health. However, self-perceptions often overestimate knowledge and performance [22,29,30,31]. The differences between subjective perception and objective performance, or clinical competence, should be further investigated. In health care education, it is common to assess competence via simulations with standardized patients, observed clinical performance during real-time patient encounters, and observed structured clinical exams. With standardized definitions for variables of interest, such as providing culturally competent care or promoting a healthy lifestyle, these assessments could be utilized in future research to verify the degree to which clinicians are implementing the principles of PCC and the BPS model of health in their clinical practice. Additionally, the items of the PCC and BPS questionnaires are vague enough that participants could have responded to items with what they believe is socially desirable. In future studies, it may be necessary to provide concrete examples of specific behaviors that represent providing culturally competent care, promoting a healthy lifestyle, etc. 

Further, a potential limitation of the study is the low response rate. A low response rate could lead to sampling bias and may not be representative of the sample within the target population.

## 5. Conclusions

The results of the present study suggest that SSATs perceive they are integrating the principles of PCC and the BPS model of health in clinical practice. These findings align with two previous studies [1,4] that concluded collegiate student-athletes and parents of minor student-athletes perceive athletic trainers as providing PCC. These collective findings support the conclusion that patients, parents, and providers believe athletic trainers are providing care that is patient-centered and focused on whole-person healthcare. However, previous studies have shown that individuals rate themselves with higher perceptions of care than actual knowledge [22]. The difference between self-perception and actual performance could explain why these studies have concluded that athletic trainers are providing PCC because they have assessed perception and not objective performance. 

SSATs can improve their PCC by including a validated, focused history script, utilizing health informatics, and taking a multidimensional approach to incorporating these concepts. In doing so, athletic trainers can better identify specific patient needs and begin to improve patient outcomes. The goal of providing intentional, whole-person healthcare to patients is to understand the determinants of injury and illness in order to arrive at effective treatments and patterns of healthcare. To do that, athletic trainers must consider the patient, the social context in which they operate, and the complementary system built to address the condition (i.e., the healthcare provider role and the healthcare system) [15]. In doing so, athletic trainers can gain a deeper understanding of patients and their health outcomes and work towards a more realistic approach to care [12]. 

## Figures and Tables

**Table 1 ijerph-20-05480-t001:** The Picker Institute’s Principles of Patient-Centered Care.

Harvey Picker’s Eight Domains of PCC
Respect for patients’ preferences Coordination and integration of care Information and education Physical comfort Emotional support Involvement of family and friends Continuity and transition Access to care

**Table 2 ijerph-20-05480-t002:** Participant Characteristics.

Characteristic	
Age, mean ± SD	37.8 ± 12.6
Gender, *n* (%)	
Man	133 (37.9)
Woman	215 (61.3)
Non-Binary/Gender Nonconforming	1 (0.3)
Transgender Woman	1 (0.3)
Prefer Not to Say	1 (0.3)
Current Position, *n* (%)	
Full time secondary school athletic trainer	249 (70.9)
Part time secondary school athletic trainer	27 (7.7)
Split clinic athletic trainer and secondary school athletic trainer	25 (7.1)
Split educator and secondary school athletic trainer	46 (13.1)
Other	4 (1.1)
Highest Degree Earned, *n* (%)	
Bachelor’s (BA, BS, etc.)	102 (29.1)
Master’s (MA, MS, etc.)	234 (66.7)
Academic Doctorate (PhD, EdD, DHSc, etc.)	5 (1.4)
Clinical Doctorate (DAT, DPT, etc.)	9 (2.6)
Other	1 (0.3)
Years BOC Certified, mean ± SD	13.8 ± 11.6
Years Clinically Practicing, mean ± SD	13.5 ± 11.5
Years Employed in Current Position, mean ± SD	8.2 ± 8.9

## Data Availability

Due to the nature of the questions asked in this study, survey respondents were assured that raw data would remain confidential and would not be shared.

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
