# Peer review of "The Integration of Patient-Centered Care and the Biopsychosocial Model by Athletic Trainers in the Secondary School Setting"

_ijerph, 2023, doi:10.3390/ijerph20085480_

Round 1

Reviewer 1 Report

This is a nice study that adds to the body of literature. I am a little confused why a journal more specific to athletic training was not considered, as this is important information for that population specifically. The tools are appropriately validated.

Table 2:  I would argue the DAT is an advanced practice doctorate (post-professional), more in line with a DNP, whereas a DPT is a professional, entry level doctorate. I am not sure combining those two are appropriate. Since the data is not analyzed by educational level, it is likely not problematic. Something to consider in the future.

References: Citations 10 and 20 appear to be the same citation.

Author Response

  • This is a nice study that adds to the body of literature. I am a little confused why a journal more specific to athletic training was not considered, as this is important information for that population specifically. The tools are appropriately validated.
    • Thank you for reviewing the paper. This journal was submitted for the special edition focuses on patient-centered care in sports medicine services. While the paper specifically focuses on athletic trainers in the secondary school setting, the tools used in this study could be used for any sports medicine provider working with patients in the secondary school, including physical therapists, occupational therapists, and school nurses. Providers working in the secondary school setting must consider many factors affecting patients, including the social context in which they operate and the system built to address their needs. We believe that this manuscript is appropriate for publication by the International Journal of Environmental Research and Public Health because it explores the extent to which secondary school athletic trainers provide patient-centered care and discusses strategies for implementation that can have positive impact to all providers in the space.
  • Table 2:  I would argue the DAT is an advanced practice doctorate (post-professional), more in line with a DNP, whereas a DPT is a professional, entry level doctorate. I am not sure combining those two are appropriate. Since the data is not analyzed by educational level, it is likely not problematic. Something to consider in the future.
    • Thank you for the commentary. I understand the argument for not combining DAT degree with DPT degree within the demographics section, and we agree that this is not likely problematic in the context for this study. The degrees were grouped based on the criteria of conferring a doctoral degree in a clinical space as opposed to an academic or research doctorate. However, the implications of newly practicing versus currently practicing clinicians would be something to consider. We appreciate the suggestion and will most certainly consider this in the future.
  • References: Citations 10 and 20 appear to be the same citation.
    • Thank you. Citations 10 and 20 were revised as suggested.

Reviewer 2 Report

The Integration of Patient-Centered Care and the Biopsychosocial Model by Athletic Trainers in the Secondary School Setting

This paper deals with the degree to which secondary school athletic trainers (SSATs) perceive they are integrating the principles of patient-centered care (PCC) and the biopsychosocial (BPS) model in their practice.

Abstract, title and references         

Abstract is concise and clearly written, with a good command of English, and clear representation of the aim of the paper. Containing 200 words, it does meet the demands of the journal IJERPH (200 max). Furthermore, it is adequately structured: background of the proposed research, analytical methods used, and main conclusions were mentioned.

The title of the paper adequately reflects the subject under investigation in the proposed study.

References are numbered in order of appearance in the text, as demanded by formatting rules of the journal. Although there is no limitation in the number of references, a reference list of 30 citations is completely sufficient to cover the topic proposed.

Introduction 

The authors clearly represented the importance of the issue described. Please insert the list of abbrevations.

Line 31: Please explain the previous level of knowledge of the atletic trainers regarding PCC and BPS, or later in the text related on atletic trainers sample in part 2.2. is there some official education on the state level.

Materials and methods

Line 66: Please specify which national assocation

Line 105: full name of abbreviations

Results and discussion

Authors used adequate tools and statistics in order to characterize the colected date. As support for this research, it is advisable to analyze the data in the light of gender issues, i.e. whether there are significant differences between female and male athletic trainers in percieving of integration these to prnciples, PCC and BPS,  in their work.

The Integration of Patient-Centered Care and the Biopsychosocial Model by Athletic Trainers in the Secondary School Setting

This paper deals with the degree to which secondary school athletic trainers (SSATs) perceive they are integrating the principles of patient-centered care (PCC) and the biopsychosocial (BPS) model in their practice.

Abstract, title and references         

Abstract is concise and clearly written, with a good command of English, and clear representation of the aim of the paper. Containing 200 words, it does meet the demands of the journal IJERPH (200 max). Furthermore, it is adequately structured: background of the proposed research, analytical methods used, and main conclusions were mentioned.

The title of the paper adequately reflects the subject under investigation in the proposed study.

References are numbered in order of appearance in the text, as demanded by formatting rules of the journal. Although there is no limitation in the number of references, a reference list of 30 citations is completely sufficient to cover the topic proposed.

Introduction 

The authors clearly represented the importance of the issue described. Please insert the list of abbrevations.

Line 31: Please explain the previous level of knowledge of the atletic trainers regarding PCC and BPS, or later in the text related on atletic trainers sample in part 2.2. is there some official education on the state level.

Materials and methods

Line 66: Please specify which national assocation

Line 105: full name of abbreviations

Results and discussion

Authors used adequate tools and statistics in order to characterize the colected date. As support for this research, it is advisable to analyze the data in the light of gender issues, i.e. whether there are significant differences between female and male athletic trainers in percieving of integration these to prnciples, PCC and BPS,  in their work.

Author Response

  • The authors clearly represented the importance of the issue described. Please insert the list of abbreviations.
    • Thank you for this comment. It is unclear what the reviewer is indicating with the “list of abbreviations,” as these are not required on the author submission guidelines.
  • Line 31: Please explain the previous level of knowledge of the athletic trainers regarding PCC and BPS, or later in the text related on athletic trainers sample in part 2.2. is there some official education on the state level.
    • Thank you for the suggestion. Athletic training education is guided by the curricular standards set by the Commission on Accreditation of Athletic Training Education (CAATE). This is a national organization, and these standards apply to all athletic training programs in the United States. This section has been revised for clarity, and information regarding the CAATE standards and prior athletic trainer education on PCC and BPS models of health is provided in lines 175-183
  • Line 66Please specify which national association
    • Thank you for the commentary. This is provided in line 68 “National Athletic Training Association.”
  • Line 105: full name of abbreviations
    • Thank you. This has been revised as suggested.
  • Authors used adequate tools and statistics in order to characterize the collected date. As support for this research, it is advisable to analyze the data in the light of gender issues, i.e. whether there are significant differences between female and male athletic trainers in perceiving of integration these to principles, PCC and BPS, in their work.
    • Thank you for the suggestion. We have included this analysis.

Reviewer 3 Report

The purpose of this study, as stated at the end of the INTRODUCTION, is to investigate the extent to which secondary school athletic trainers (SSATs) perceive the integration of principles of patient-centered care (PCC) principles and the biopsychosocial (BPS) model into their practice. However, it is unfortunate that the hypothesis of this study is not stated at all and remains a simple survey.

In this paper, the authors should consider and clearly state what is academically important and scientifically evident.

Author Response

  • The purpose of this study, as stated at the end of the INTRODUCTION, is to investigate the extent to which secondary school athletic trainers (SSATs) perceive the integration of principles of patient-centered care (PCC) principles and the biopsychosocial (BPS) model into their practice. However, it is unfortunate that the hypothesis of this study is not stated at all and remains a simple survey.
    • Although hypotheses are not common in exploratory research, we have included one.
  • In this paper, the authors should consider and clearly state what is academically important and scientifically evident.
    • Thank you for the suggestion. The findings are clearly communicated in the results and then discussed, in comparison to available literature.

Reviewer 4 Report

Results section 3.1 is a bit confusing. Recommend rephrasing for better understanding. 

Minor English edits needed. 
Ie Line 41 should be "leads" and Line 221 should be affects (no ')

Otherwise, well done.

Author Response

  • Results section 3.1 is a bit confusing. Recommend rephrasing for better understanding. 
    • Thank you. Section 3.1 was revised as suggested.
  • Minor English edits needed. Ie Line 41 should be "leads" and Line 221 should be affects (no ')
    • Lines 41 and 221 were revised as suggested. Please note that we have also used Grammarly to review the entire manuscript for English edits with minor edits made throughout the manuscript for clarity.

Round 2

Reviewer 3 Report

As the authors state, the academic importance of this paper is well understood. However, the authors need to be more careful in stating what the novelty in this paper is with respect to previous studies.

Author Response

  • As the authors state, the academic importance of this paper is well understood. However, the authors need to be more careful in stating what the novelty in this paper is with respect to previous studies.
    • Thank you for this comment. We agree that this approach is not novel from a medical research perspective. After reviewing the document, we have made sure that we are not improperly identifying our research as generally novel. However, we do believe that this approach is novel in research within athletic training, and we have added in specific language reflecting this belief on lines 55-57.